# Is Glyceryl Trinitrate, a Nitric Oxide Donor Responsible for Ameliorating the Chemical-Induced Tissue Injury In Vivo?

**DOI:** 10.3390/molecules27144362

**Published:** 2022-07-07

**Authors:** Ayesha Rahman Ahmed, Mahiba Ahmed, Senty Vun-Sang, Mohammad Iqbal

**Affiliations:** 1Department of Medical Elementology and Toxicology, Faculty of Science, Hamdard University, New Delhi 110062, India; ayesha.ahmed@wsu.edu; 2Voiland School of Chemical Engineering and Bioengineering, Pullman, WA 99164, USA; mahiba.ahmed@wsu.edu; 3Biotechnology Research Institute, Universiti Malaysia Sabah, Jalan UMS, Kota Kinabalu 88400, Sabah, Malaysia; sentyvunsang2707@gmail.com

**Keywords:** tissue injury, nitric oxide, glyceryl trinitrate, NG-nitro-l-arginine methyl ester, oxidative stress, organ toxicity

## Abstract

Oxidative stress induced by well-known toxins including ferric nitrilotriacetate (Fe-NTA), carbon tetrachloride (CCl_4_) and thioacetamide (TAA) has been attributed to causing tissue injury in the liver and kidney. In this study, the effect of glyceryl trinitrate (GTN), a donor of nitric oxide and NG-nitroarginine methyl ester (l-NAME), a nitric oxide inhibitor on TAA-induced hepatic oxidative stress, GSH and GSH-dependent enzymes, serum transaminases and tumor promotion markers such as ornithine decarboxylase (ODC) activity and [^3^H]-thymidine incorporation in rats were examined. The animals were divided into seven groups consisting of six healthy rats per group. The six rats were injected intraperitoneally with TAA to evaluate its toxic effect, improvement in its toxic effect if any, or worsening in its toxic effect if any, when given in combination with GTN or l-NAME. The single necrogenic dose of TAA administration caused a significant change in the levels of both hepatic and serum enzymes such as glutathione S-transferase (GST), glutathione reductase (GR), glutathione peroxidase (GPx), γ-glutamyl transpeptidase (GGT), glucose 6-phosphate dehydrogenase (G6PD), alanine aminotransferase (AST) and aspartate aminotransferase (ALT). In addition, treatment with TAA also augmented malondialdehyde (MDA), ornithine decarboxylase (ODC) activity and [^3^H]-thymidine incorporation in rats liver. Concomitantly, TAA treatment depleted the levels of GSH. However, most of these changes were alleviated by the treatment of animals with GTN dose-dependently. The protective effect of GTN against TAA was also confirmed histopathologically. The present data confirmed our earlier findings with other oxidants including Fe-NTA and CCl_4_. The GTN showed no change whatsoever when administered alone, however when it was given along with TAA then it showed protection thereby contributing towards defending the role against oxidants-induced organ toxicity. Overall, GTN may contribute to protection against TAA-induced oxidative stress, toxicity, and proliferative response in the liver, according to our findings.

## 1. Introduction

The liver being a vital organ of the human body is responsible for the detoxification of xenobiotics and maintenance of metabolic functions. The therapy against toxic agents could be hindered due to their major involvement in the body’s cellular modification. Various toxicants have been used as the model to cause hepatic cell injury both in the different cell types and in experimental animals [1,2,3,4]. The reactive oxygen species (ROS) has been widely studied as it participates in the etiology of various disease states and pathophysiological conditions [5,6].

One such toxicant is thioacetamide (TAA) which is known for its extensive toxicity to the hepatic system. It has wide use in the oil industry, rubber industry, and as a solvent and fungicide [7]. TAA induces acute hepatic injury that is evident by necrosis and regeneration of the hepatic cell [8,9,10,11]. The liver injury induced by TAA is widely studied. TAA has been shown to function by generating free radical intermediates through oxidative pathways that cause glutathione (GSH) depletion, lipid peroxidation [4,11,12] enhanced DNA synthesis [13], and alteration in redox cycling [8].

The protection of cellular homeostasis is the major role of the antioxidant defense system. It protects against disruption by ROS [14]. The successful functioning of the defense mechanisms includes the enzymes coordinated action to halt the cell’s pro-oxidant state and to restore the damage to the molecules to protect the cell integrity [11]. Nitric oxide (NO) is formed from the l-arginine amino acid by the enzyme nitric oxide synthase [15]. The role of NO in the human body includes regulating events of inflammation, metabolism, platelet aggregation, and vascular tone [16,17]. Despite this, multiple experimental evidence indicates NO engaging with ROS to create more toxic species [18,19], however, few studies have indicated its antioxidant role. NO, along with the radicals such as the superoxide anions could generate oxidants that are known to be less toxic [20,21]. Few published studies suggest that NO production endogenously or its administration exogenously could reduce the hepatic toxicity induced by the oxidants [3,22,23,24,25] while endogenous NO production inhibition has potentiated cellular damage [26,27]. In addition, many studies have suggested that NO is required for the daily operations of the body. NO owing to its physical and chemical properties in the human system has a diverse role in the body. Recent years have produced significant evidence indicating the NO safe impact on hepatic toxicant-induced damage to the tissue [3,27,28]. In the intervening decade since the discovery of this molecule, the NO role is intriguing and is undefined. Based on the previous findings, the objective of this research was to study the potential effect of glyceryl trinitrate (GTN), a donor of nitric oxide and NG-nitroarginine methyl ester (l-NAME), a nitric oxide inhibitor on acute liver damage and tumor promotion markers such as ornithine decarboxylase (ODC) activity and [^3^H]-thymidine incorporation in rats induced by a single necrogenic dose of TAA. GTN is a strong vasorelaxant and NO donor and is degraded to mono-and di-nitrate isomers [29,30].

## 2. Results

### 2.1. Effect of Glyceryl Trinitrate (GTN) and NG-Nitroarginine Methyl Ester (l-NAME) on Hepatic GSH-Dependent Enzymes in Rats Treated with Thioacetamide (TAA)

Table 1 shows the effect of GTN and l-NAME on hepatic GSH-dependent enzymes in rats treated with TAA. The administration of TAA resulted in a significant (*p* < 0.001) increase in glutathione S-transferase (GST), glutathione reductase (GR), glucose 6-phosphate dehydrogenase (G6PD), and γ-glutamyl transpeptidase (GGT) activities, with a concomitant decrease in glutathione peroxidase (GPx) activity compared to the group treated with saline. A significant dose-dependent reduction in GST, GR, G6PD, and GGT activities resulted from subsequent administration of GTN at two different doses. Furthermore, antioxidant enzyme activity, GPx also showed partial dose-dependent recovery to the normal value (*p* < 0.001). Administration of l-NAME, a nitric oxide inhibitor further enhanced TAA-induced oxidative stress by increasing GST, GPx, and G6PD with a simultaneous reduction in GPx activity.

### 2.2. Effect of Glyceryl Trinitrate (GTN) and NG-Nitroarginine Methyl Ester (l-NAME) on Serum Transaminases in Rats Treated with Thioacetamide (TAA)

Table 2 shows the effect of GTN and l-NAME on serum transaminases in rats treated with TAA. A significant (*p* < 0.001) enhancement of serum transaminases alanine aminotransferase (AST) and aspartate aminotransferase (ALT) marker enzymes of liver damage were noticed with TAA administration. When compared to the TAA treatment group, the administration of GTN at doses of 3 and 6 mg/kg body weight resulted in substantial inhibition of the release of serum transaminases (*p* < 0.001). However, l-NAME administration to TAA-intoxicated rats further enhanced the release of transaminases AST and ALT in serum (*p* < 0.01).

### 2.3. Effect of GTN and l-NAME on Hepatic GSH and Lipid Peroxidation in Rats Treated with Thioacetamide (TAA)

Figure 1a,b show the effect of GTN and l-NAME on hepatic GSH (Figure 1a) and lipid peroxidation (Figure 1b) in rats treated with TAA. TAA administration at a dose of 500 mg/kg bwt, resulted in a significant (*p* < 0.01) depletion of GSH content as compared to the saline-treated control group (Figure 1a). This was accompanied by a corresponding rise in lipid peroxidation (LPO) significantly (*p* < 0.001) as compared to the saline-treated control group (Figure 1b). When compared to the group only treated with TAA, the administration of GTN at doses 3 and 6 mg/kg bwt, showed a significant (*p* < 0.05) dose-dependent increase in the content of GSH and inhibition of the development of thiobarbituric acid (TBA) reacting species (*p* < 0.01). l-NAME treatment after TAA caused further depletion of GSH and elevations in LPO were noticed. GTN and l-NAME alone did not cause any major changes in those parameters.

### 2.4. Effect of GTN and l-NAME on Hepatic Ornithine Decarboxylase (ODC) Activity and [^3^H]-Thymidine Incorporation in Rats Treated with Thioacetamide (TAA)

Figure 2a,b show the effect of GTN and l-NAME on hepatic ODC activity (Figure 2a) and [^3^H]-thymidine incorporation (Figure 2b) in rats treated with TAA. TAA administration resulted in a significant enhancement of hepatic ODC activity and [^3^H]-thymidine incorporation as compared to their respective saline-treated control groups (*p* < 0.001). A dose-dependent inhibition of hepatic ODC activity (*p* < 0.001) and [^3^H]-thymidine incorporation into the hepatic DNA (*p* < 0.01) resulted from subsequent administration of GTN suggesting that it may contribute to protection against TAA-induced proliferative response. l-NAME increased TAA’s proliferative action resulting in a further increase in hepatic ODC activity (*p* < 0.01) and [^3^H]-thymidine incorporation relative to the TAA group.

### 2.5. Effect of GTN and l-NAME on Hepatic Histopathology in Rats Treated with Thioacetamide (TAA)

Figure 3 shows the effect of GTN and l-NAME on hepatic histopathology in rats treated with TAA. TAA administration caused the increased infiltration of the inflammatory cellular components around the central vein and portal triad resulting in the degeneration of 5–6 layers of hepatocytes around blood vessels (Figure 3D). These changes were restored on subsequent administration of GTN dose-dependently, as evident from mild grade inflammatory cellular infiltrates and degeneration of 3–4 layers of hepatocytes (Figure 3E). Dilated portal vein of the larger lumen and hepatic artery of the smaller lumen is seen in the portal triad. However, GTN and l-NAME alone administration showed no pathological changes (Figure 3B,C). l-NAME administration in TAA-treated rats further enhanced TAA-mediated pathological changes showing an enhanced number of inflammatory cells and moderate infiltration leading to a degeneration of hepatocytes around the portal triad (Figure 3F).

### 2.6. Amount of Nitrite Generated by GTN

The maximum amount of nitrite detected was found to be 0.75 ± 0.2 and 1.7 ± 0.3 mM/g tissue, at 3 and 6 mg doses of GTN, following incubation of GTN with hepatic PMS in vitro for a total of 45 min.

## 3. Discussion

Several studies were performed in the past to better understand the role of NO as a signaling molecule, a pro-oxidant, and a potent antioxidant. Due to its contradictory chemistry and biologic activity in oxidative stress, it can play a dual role as a pro-toxicant or as an antioxidant that depends on balance shift at its high concentration. In this study, we sought to evaluate the role of NO donor and inhibitor in TAA-induced liver toxicity and tumor promotion markers in rats. The results of the current research confirm that TAA induces liver toxicity that could be seen with the decrease in NO level. However, on the positive side, oxidative stress and cell proliferation in the hepatic system could be protected by nitric oxide donor at the concentration where it maintains the redox balance of the cell.

The pathophysiological role of the TAA and its characterization in causing hepatic toxicity has been well defined in this study. Liver injury is well known to be caused by TAA through oxidative stress and proliferation of the hepatic cell. TAA is known to be an intense toxicant containing thiono-sulfur which not only inflicts hepatic injury but also acts as an oxidant to cause cellular stress and proliferative response from ROS. The findings of this study strongly suggest that NO generated through the administration of GTN has hepatoprotective action in acute TAA-induced injury. This effect of GTN may be due to the generation of a relatively fewer toxic species with reactive metabolites and other free radicals formed as a direct consequence of bioactivation with TAA [3,22].

TAA-induced hepatic damage is an emerging field of pharmacological interest as free radicals produced by the drug oxidation of microsomes relate to the mechanism of programmed cell death [11]. The intense lipid peroxidation and the depletion of the content of tissue glutathione (GSH) have been suggested as a marker of the TAA hepatic toxicity [10], leading to instability in the GSH redox cycle and eventually oxidant stress [13]. Accordingly, on subsequent administration of GTN, the depletion in the glutathione and increase in the lipid peroxidation were significantly reversed depending on the dose of GTN. NOS inhibitor administration, l-NAME, causes additive toxicity as could be seen by the GSH depletion and elevation of lipid peroxidation. GTN exerts the protection against liver injury which was evident by the amelioration of the enzyme levels such as ALT, GGT and AST. NO is demonstrated to inhibit oxidant generation [31] and terminate the lipid peroxidation reactions in the various lipid systems [31,32] in different models [33,34] and in cells [24,35] therefore, subsequent administration of GTN could be suggested to suppress the bioactivation of TAA, demonstrating its healing effect by the decline in the GSH conjugation-reactive metabolites and lowering the damage inflicted by ROS. Our findings also concur with previous research involving NO inhibition as an additive cause of hepatic injury [3,21,24].

The glutathione redox cycle plays an important role in keeping the cell’s thiol oxidative balance [36,37]. Depletion of GSH has been shown to cause impaired oxidative defenses and demonstrated a disturbance of vital biochemical functions [38]. It has been shown previously that the regulation of radical hydroxyl levels involves catalyzing glutathione-S-transferase (GST) electrophilic conjugation of reactive intermediates with glutathione [39,40] and glutathione peroxidase (GPx). Meanwhile, glutathione reductase (GR) uses GSSG and refills GSH using NADPH as a source of energy [41]. The significant stimulation of various antioxidant enzymes such as glutathione-S-transferase, glutathione reductase, and glutathione-6-phosphate dehydrogenase activities is a result of TAA administration with a sequential decline in glutathione peroxidase function, which could be as a result of TAA-induced persistent oxidative stress signaling the initiation of genes that are accountable for antioxidant enzyme countenance. Glucose 6-phosphate dehydrogenase, although not directly in GSH metabolism, provides NADPH needed for reduction of oxidized glutathione. The activities of all these enzymes determine the tissue levels of GSH. The improvement in TAA induced oxidative stress due to GTN, a prodrug generating NO could possibly be attributed to its ability to scavenge the lipid peroxide radical, while suppressing the level of oxidative stress. The attenuation of GSH and GSH dependent enzymes in TAA intoxicated rats by GTN could also result from the competitive blockage of nucleophile sites to outclass electrophilic molecules, thereby reducing the use of GSH enzymes to their partial normal values and the concomitant recovery of antioxidant enzymes. GTN was helpful in protecting hepatic cells against TAA induced oxidative injury as observed in this study.

In response to the xenobiotic attack, the liver that was regenerating post-injury provides a model of cell proliferation to research the cause of the cell division modulation [42,43,44]. After cell loss in hepatotoxins induced tissue injury, the liver of mammals has been shown to possess the capacity to grow [45,46]. TAA’s induced cell death is immediately accompanied by proliferative response reaching a height of 48 h post-TAA intoxication [43]. It has been shown that intracellular GSH concentrations affect DNA either by modulating DNA synthesis [47] or by protecting DNA from oxidative damage [37,48]. Increased ODC activity leads to an enhanced rate of DNA synthesis that correlates with tumor promoting potency of an agent [5,49]. TAA treatment resulted in a significant enhancement of ODC activity and [^3^H]-thymidine incorporation. GTN administration following TAA intoxication resulted in a significant down regulation of ODC activity and [^3^H]-thymidine incorporation. The mechanism of protection of GTN against TAA-induced hepatic ODC activity and [^3^H]-thymidine incorporation may be attributed to its ability to decrease oxidant production by inflammatory cells. Overall, in this study our results suggest that NO donor GTN can protect against TAA induced proliferative response in the liver.

TAA is a selective hepatotoxicant as its metabolism produces a highly reactive compound, TAA-S oxide that binds to the tissue macromolecules which can lead to hepatic necrosis [50]. The induction of GGT is a commonly used marker for pre-neoplastic liver lesions in rats due to enhanced oxidative stress. In our study, the serum enzyme tests indicated an elevation in ALT, GGT and AST levels after TAA treatments possibly be due to plasma membrane damage produced by LPO which were reversed after administration of GTN. Previous studies emphasized the existence of liver failure due to TAA administration leading to an enhanced level of liver enzymes (AST, GGT and ALT), oxidative stress manifested by increased MDA levels with reduction of GSH and NO levels in liver tissue homogenate [28,51,52]. Thus, oxidative stress induced by TAA is explained previously to be the effect of a decrease in NO, reduced GSH, and an increase in MDA level and serum enzymes. Our study supports this model as l-NAME enhances the level of ALT, GGT, and AST, while the significant reduction of liver enzymes ALT, GGT, and AST by GTN as a NO donor confirms its role as an antioxidant and points towards a reduced TAA-induced hepatocellular necrosis, according to our findings.

## 4. Materials and Methods

### 4.1. Chemicals ad Reagents

Thioacetamide, thiobarbituric acid, 2,4-dinitrophenylhydrazine, NG-nitroarginine methyl ester (l-NAME), glyceryl trinitrate, horseradish peroxidase, ethylene diamine tetra-acetic acid (EDTA), phenol red, nicotinamide adenine dinucleotide phosphate (NADP), tris-HCl, 1-chloro-2,4-dinitrobenzene (CDNB), dithiothreitol, oxidized and reduced glutathione (GSH), glucose-6-phosphate, 2-mercaptoethanol, pyridoxal-5-phosphate, phenylmethylsulfonyl fluoride (PMSF), 5,5-dithio-bis-2-nitrobenzoic acid (DTNB) and γ-glutamyl-*p*-nitroanilide were purchased from Sigma (St. Louis, MO, USA). Sulfosalicylic acid, naphthalene, 1,4-bis (5-phenyl-2-oxazolyl) benzene, 1,4-dioxane, toluene, perchloric acid (PCA), sodium chloride, potassium chloride, trichloroacetic acid (TCA), ascorbic acid, citric acid and sodium hydroxide (NaOH) from Central Drug House (P) Ltd (CDH), New Delhi, India.. 2,5-Diphenyloxazole and ferric chloride were purchased from Spectrochem Private Limited, Mumbai, India. Methanol, tween-80, brij-25 and sodium azide were purchased from SD Fine Chem Limited, Mumbai, India. [^14^C]ornithine (specific activity 82 Ci/mmol) and [^3^H]-thymidine (specific activity 56 m Ci/mmol) have been obtained from Amersham (Buckinghamshire, UK). All other chemicals and reagents that were used were of the highest purity commercially available.

### 4.2. Thioacetamide (TAA) Preparation

TAA solution was prepared as per the earlier method [53]. The TAA solution was freshly diluted in normal saline (0.9% *w*/*v* NaCl) immediately before its use.

### 4.3. Scintillation Fluid Preparation

1,4-bis(5-phenyl-2-oxazolyl) benzene (0.065 g), 2,5-diphenyloxazole (3.25 g), naphthalene (52 g) have been dissolved in a mixture containing methanol (150 mL), toluene (aldehyde free, 250 mL) and 1,4-dioxane (250 mL) and placed in a dark bottle.

### 4.4. Animals and Experimental Protocols

The approval of the animal experiments was received by the Ethical Animal Care Committee of Jamia Hamdard, Delhi, India (JH AEC-03/1997). The animals were kept according to the guidelines of the Indian Council of Medical Research (ICMR). Throughout this study, the age range selected was 4–6 weeks old for the Wistar strain male albino rats with a weight range of 125–150 g. The rats were kept at Jamia Hamdard Central Animal House Colony in six rats grouped per cage in polypropylene cages. The rat’s cage was housed in a room maintained at a temperature that was considered normal a 12 h light and dark period and acclimatized for one week. The rats were provided normal laboratory feed (Hindustan Lever Ltd., Bombay, India) and tap water ad libitum. They were administered intraperitoneally with a single necrogenic dose of TAA (500 mg/kg bwt) based on previously published data and it was prepared fresh by dissolving TAA in 0.9% *w*/*v* NaCl. The TAA dose was chosen as the highest dose with survival of > 90% (8, 11). For biochemical, serum parameters, and ODC activity modulation studies, forty-two male Wistar rats were used. These rats were divided randomly into seven groups with six animals per group (Table 3).

After treatment with TAA/saline, all the animals were sacrificed after 24 h by cervical dislocation. Blood was taken by posterior vena cava before the heart stopped beating and the serum was separated which was used for serum transaminase activity estimation. The liver of the rats was also extracted. The liver has been rinsed with ice-cold saline (0.9% *w*/*v* NaCl) for biochemical and ODC activity modulation studies.

For the [^3^H]-thymidine incorporation study, there were additional forty-two male Wistar rats were used. A similar experimental protocol was followed as mentioned above for the [^3^H]-thymidine incorporation study, with the exception that each rat was given [^3^H]-thymidine (30 μCi/animal/0.2 mL saline) 2 h before killing. The liver parts were removed, rinsed with ice-cold saline (0.9% *w*/*v* NaCl), and then processed into hepatic DNA for quantification of [^3^H]-thymidine.

### 4.5. Post-Mitochondrial Supernatant (PMS, Cytosol, and Microsome Preparation)

The liver was immediately excised, infused with chilled saline (0.9% *w*/*v* NaCl), and homogenized with a Potter Elvehjem homogenizer (Omni International, Kennesaw, GA, USA) in a chilled phosphate buffer (0.1 M, pH 7.4) containing potassium chloride (KCl) (1.17% *w*/*v*). In an Eltek refrigerated centrifuge (Elektrocraft Pvt. Ltd. Mumbai, India), the homogenate was filtered and centrifuged at 800× *g* for 5 min at 4 °C to isolate the nuclear debris. The collected aliquot was centrifuged at 10,500× *g* at 4 °C for 20 min to obtain PMS, which was used as an enzyme source. A portion of the PMS was centrifuged for 60 min at 4 °C, 105,000× *g* in an ultracentrifuge, and the obtained cytosol and pellet, known as microsomal fraction, was suspended in phosphate buffer (0.1 M, pH 7.4, containing (1.17% *w*/*v* KCl).

### 4.6. Reduced Glutathione (GSH)

GSH in the liver PMS was tested using the method described previously [54]. The yellow color developed was read at 412 nm using a spectrophotometer.

### 4.7. Glutathione S-transferase (GST)

GST activity was measured in hepatic PMS using the method described previously [55]. The variations in the absorbance were analyzed at 340 nm and the activity of the enzyme was measured as nmol CDNB conjugate formed/min/mg protein using a 9.63 × 10^3^ M/cm molar extinction coefficient.

### 4.8. Glutathione Reductase (GR)

GR activity in hepatic PMS was investigated using the stated method by Gnanaraj et al. [55]. The enzyme activity was measured by the NADPH disappearance at 340 nm at 25 °C and calculated using a molar extinction coefficient of 6.223 × 10^3^ M/cm as nmol NADPH oxidized/min/mg protein.

### 4.9. Glutathione Peroxidase (GPx)

The activity of GPx in hepatic PMS was measured using the procedure described in Gnanaraj et al. [55]. NADPH disappearance was reported for 3 min at 30-s intervals at 340 nm at 25 °C. The function of the enzyme was determined using a molar extinction coefficient of 6.22 × 10^3^ M/cm as nmol NADPH oxidized/min/mg protein.

### 4.10. γ-glutamyl Transpeptidase (GGT)

A substrate y-glutamyl *p*-nitroanilide in hepatic PMS was used to determine the GGT activity following the Tamura et al. method [56]. The activity of the enzyme was read at 405 nm and determined using a molar extinction coefficient of *p*-nitroaniline as 1.74 × 10^3^ M/cm as the nmol *p*-nitroaniline formed/min/mg protein.

### 4.11. Glucose 6-phosphate Dehydrogenase (G6PD)

G6PD activity in hepatic PMS was assessed using the method used by Shah et al. [57]. The absorbance variations were reported at 340 nm and the enzyme activity was measured using a molar extinction coefficient of 6.223 × 10^3^ M/cm as a reduced/min/mg protein nmol NADP.

### 4.12. Lipid Peroxidation

The hepatic microsomal lipid peroxidation assay was conducted using the method described by Gnanaraj et al. [58]. The amount of malondialdehyde produced in each of the samples was measured using a spectrophotometer to measure the supernatant’s optical density at 535 nm against a blank reagent. The results were expressed by using a molar extinction coefficient of 1.56 × 10^5^ M^−1^ cm^−1^ as nmol MDA formed/h/g tissue at 37 °C.

### 4.13. Ornithine Decarboxylase (ODC) Activity

ODC activity in hepatic cytosol was calculated by measuring ^14^CO_2_ release from dl-[1-^14^C]ornithine using the method defined by Ray et al. [59]. The radioactivity was counted in a liquid scintillation counter and ODC activity was expressed as a protein released by pmol ^14^CO_2_/h/mg.

### 4.14. [^3^H]-thymidine Incorporation Assay

The amount of [^3^H]-thymidine that was incorporated into DNA was calculated using [^3^H] counting in liquid scintillation as previously mentioned [60]. As DPM/μg DNA, the amount of [^3^H]-thymidine incorporated in hepatic DNA was expressed.

### 4.15. Serum Transaminases (AST and ALT)

The assay for serum transaminase activity was carried out according to Reitman and Frankel [61]. The color developed in the reaction mixture was read at 505 nm.

### 4.16. Histopathological Studies

Mid-sections of livers measuring a few millimeters thick from each group were excised and processed for histopathological examination to support biochemical findings. The histopathological study involves fixing liver specimens in 10% neutral buffered formalin, then preparing the blocks in paraffin for microtome sectioning (5–6 µm thick), and then staining with Hematoxylin and Eosin (H&E). Finally, liver sections were examined by a pathologist under high-resolution light microscopic examination with photographic facilities.

### 4.17. Nitrite Determination

The amount of nitrite released by GTN was measured by using the method of Rahman et al. [3], as its stable oxidative metabolite, nitrite (NO_2_) in vitro. The absorbance at 540 nm was measured and the nitrite concentration was determined using sodium nitrite as a standard.

### 4.18. Protein Assay

The Lowry et al., method using bovine serum albumin as a standard was used to determine protein in all samples [62].

### 4.19. Statistical Analysis

The values were reported as mean ± standard error of the mean (SEM). The significance level between different groups is based on the *t*-test by Dunnett, followed by the analysis of variance (ANOVA). The statistical significances between different groups were calculated using One-way ANOVA in which the *p* < 0.05 value was deemed statistically significant.

## 5. Conclusions

In conclusion, present data suggest that NO generated through the administration of GTN might scavenge ROS and decrease the toxic metabolite of TAA, thereby inhibiting hepatic oxidative stress and toxicity in rats. In addition, GTN can also inhibit TAA-induced ODC activity and [^3^H]-thymidine incorporation into hepatic DNA. Overall, GTN suppresses TAA-induced oxidative stress, toxicity, and proliferative response, according to our findings. Based on the above results, the present study suggests that GTN may be a potential therapeutic agent for the restoration of TAA-induced oxidative damage and proliferative response in the liver.

## Figures and Tables

**Figure 1 molecules-27-04362-f001:**
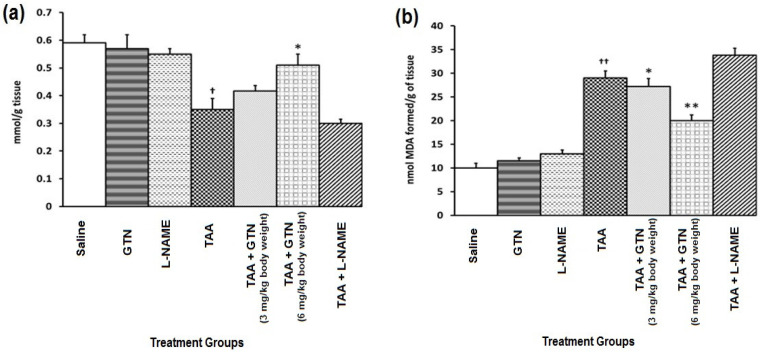
(**a**) Effect of GTN and l-NAME on hepatic GSH in rats treated with TAA. (**b**) Effect of GTN and l-NAME on lipid peroxidation in rats treated with TAA. Data represent mean ± SEM of six rats/group. The dose regimen and treatment protocol are described in the text. Group I: Saline treatment, Group II: Glyceryl trinitrate (GTN) treatment, Group III: NG-nitroarginine methyl ester (l-NAME) treatment, Group IV: Thioacetamide (TAA) treatment, Group V: TAA + GTN (3 mg/kg bwt) treatment, Group VI: TAA + GTN (6 mg/kg bwt) treatment, Group VII: TAA + l-NAME treatment. ^†^ Significantly (*p* < 0.01) different from the saline-treated group. ^††^ Significantly (*p* < 0.001) different from the saline-treated group. * Significantly (*p* < 0.05) different from the TAA-treated group. ** Significantly (*p* < 0.01) different from the TAA-treated group.

**Figure 2 molecules-27-04362-f002:**
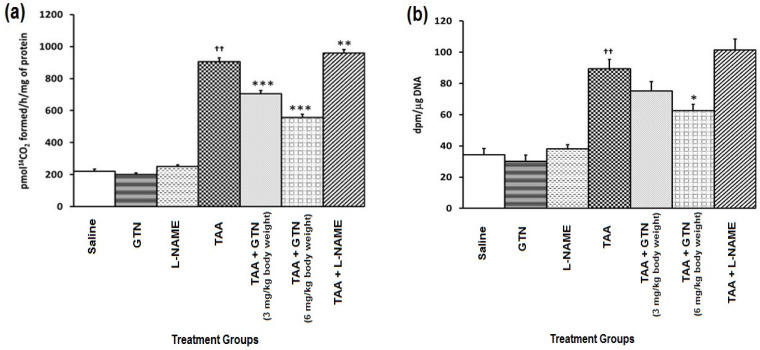
(**a**) Effect of GTN and l-NAME on hepatic ODC activity in rats treated with TAA. (**b**) Effect of GTN and l-NAME on [^3^H]-thymidine incorporation in rats treated with TAA. Data represent mean ± SEM of six rats/group. The dose regimen and treatment protocol are described in the text. Group I: Saline treatment, Group II: Glyceryl trinitrate (GTN) treatment, Group III: NG-nitroarginine methyl ester (l-NAME) treatment, Group IV: Thioacetamide (TAA) treatment, Group V: TAA + GTN (3 mg/kg bwt) treatment, Group VI: TAA + GTN (6 mg/kg bwt) treatment, Group VII: TAA + l-NAME treatment. ^††^ Significantly (*p* < 0.001) different from the saline-treated group. * Significantly (*p* < 0.05) different from the TAA-treated group. ** Significantly (*p* < 0.01) different from the TAA-treated group. *** Significantly (*p* < 0.001) different from the TAA-treated group.

**Figure 3 molecules-27-04362-f003:**
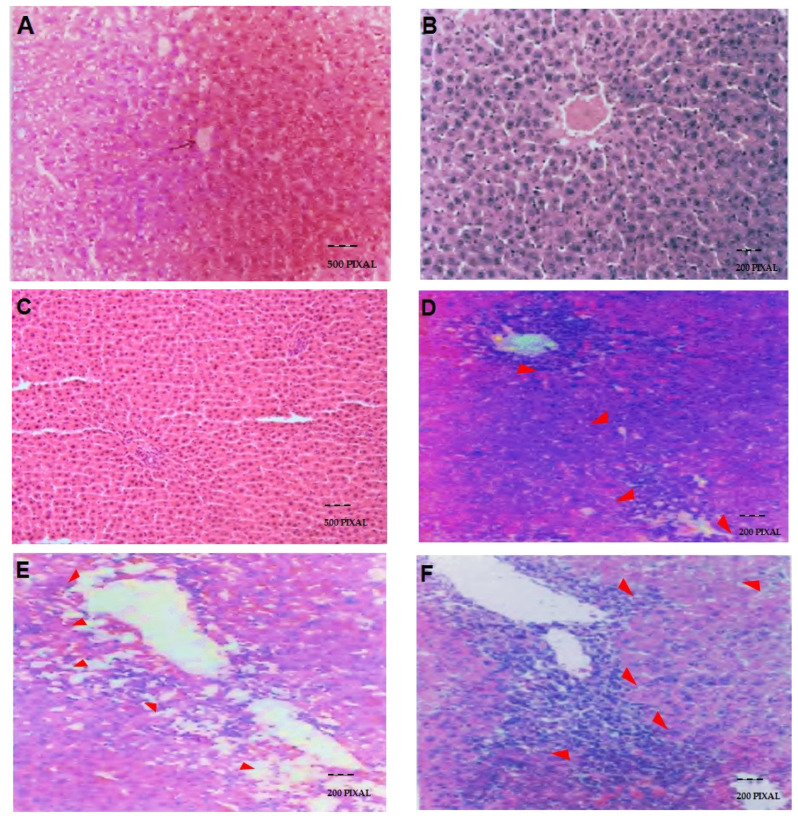
Effect of GTN and l-NAME on hepatic histopathology in rats treated with TAA. (**A**) = Saline; (**B**) = GTN; (**C**) = l-NAME; (**D**) = TAA; (**E**) = TAA + GTN (6 mg/kg bwt); (**F**) = TAA + l-NAME. TAA administration showed inflammatory cellular infiltrate around the central vein and portal triad resulting in degeneration of 5–6 layers of hepatocytes around blood vessels (Figure 3D). These changes were reverted on subsequent administration of GTN dose-dependently, as evident from mild grade inflammatory cellular infiltrates and degeneration of 3–4 layers of hepatocytes. Dilated portal vein of the larger lumen and hepatic artery of the smaller lumen is seen in the portal triad (Figure 3E). However, GTN and l-NAME alone administration showed no pathological changes (**B**,**C**). l-NAME administration in TAA-treated rats further enhanced TAA-mediated pathological changes showing an enhanced number of inflammatory cells and moderate infiltration leading to a degeneration of hepatocytes around the portal triad (**F**). The dose regimen and treatment protocol are described in the text. (**A**–**F**) ×125.

**Table 1 molecules-27-04362-t001:** Effect of GTN and l-NAME on hepatic GSH-dependent enzymes in rats treated with TAA.

Treatment Groups	GST (nmol CDNB Conjugate Formed min^−1^ mg^−1^ Protein)	GR (nmol NADPH Oxidized min^−1^ mg^−1^ Protein)	GPx (nmol NADPH Oxidized min^−1^ mg^−1^ Protein)	GGT (nmol *p*-Nitroaniline Formed min^−1^ mg^−1^ Protein)	G6PD (nmol NADPH Formed min^−1^ mg^−1^ Protein)
Saline	926.73 ± 35.91	220.69 ± 8.78	248.55 ± 10.96	417.76 ± 16.17	168.23 ± 7.20
GTN	908.21 ± 42.68	232.05 ± 10.17	259.65 ± 12.83	422.20 ± 17.96	181.02 ± 13.47
l-NAME	940.39 ± 26.12	245.57 ± 12.42	219.06 ± 9.71	449.54 ± 14.48	227.36 ± 11.45 ^†^
TAA	1504.36 ± 66.52 ^††^	380.61 ± 14.42 ^††^	129.43 ± 8.57 ^††^	1041.13 ± 37.76 ^††^	349.89 ± 16.27 ^††^
TAA + GTN (3 mg/kg bwt)	1316.14 ± 59.44	306.23 ± 12.47 **	171.32 ± 9.11 **	870.71 ± 27.97 **	292.21 ± 9.10 *
TAA + GTN (6 mg/kg bwt)	1242.39 ± 45.57 **	275.72 ± 10.12 ***	198.43 ± 7.62 ***	714.39 ± 14.36 ***	267.88 ± 10.92 **
TAA + l-NAME	1682.18 ± 79.09	408.68 ± 13.45	96.22 ± 5.98 **	1132.19 ± 23.15	391.71 ± 13.74

Data represent the mean ± SEM of six animals. The dose regimen and treatment protocol are described in the text. GST: Glutathione S-transferase; GR: Glutathione reductase; GPx: Glutathione peroxidase; GGT: Gama glutamyl transpeptidase; G6PD: Glucose 6-phosphate dehydrogenase; GTN: Glyceryl trinitrate; l-NAME: NG-nitro-l-arginine methyl ester; TAA: Thioacetamide; Dose 1: Rats were administered TAA (500 mg/kg body weight (bwt), intraperitoneally (i.p.) followed by GTN (3 mg/kg bwt, i.p.) after 1 h of TAA administration; Dose 2: Rats were administered TAA (500 mg/kg bwt, i.p.) followed by GTN (6 mg/kg bwt, i.p.) after 1 h of TAA administration. l-NAME was administered (40 mg/kg bwt, i.p.) after 1 h of TAA administration. ^†^ Significantly (*p* < 0.05) different from the saline-treated group. ^††^ Significantly (*p* < 0.001) different from the saline-treated group. * Significantly (*p* < 0.05) different from the TAA-treated group. ** Significantly (*p* < 0.01) different from the TAA-treated group. *** Significantly (*p* < 0.001) different from the TAA-treated group.

**Table 2 molecules-27-04362-t002:** Effect of GTN and l-NAME on serum transaminases in rats treated with TAA.

Treatment Groups	AST (IU/L)	ALT (IU/L)
Saline	22.83 ± 1.14	16.07 ± 0.54
GTN	21.69 ± 1.55	16.26 ± 0.63 ^†^
l-NAME	28.29 ± 1.45 ^†^	20.41 ± 0.72 ^††^
TAA	75.82 ± 2.23 ^††^	46.72 ± 1.67 ^††^
TAA + GTN (3 mg/kg bwt)	56.40 ± 1.30 ***	38.37 ± 1.25 ***
TAA + GTN (6 mg/kg bwt)	44.45 ± 1.23 ***	24.97 ± 1.09 ***
TAA + l-NAME	84.57 ± 2.08 *	56.49 ± 1.38 **

Data represent the mean ± SEM of six animals. The dose regimen and treatment protocol are described in the text. AST: Aspartate aminotransferase; ALT: Alanine aminotransferase; GTN: Glyceryl trinitrate; l-NAME: NG-nitro-l-arginine methyl ester; TAA: Thioacetamide; Dose 1: Rats were administered TAA (500 mg/kg bwt, i.p.) followed by GTN (3 mg/kg bwt, i.p.) after 1 h of TAA administration; Dose 2: Rats were administered TAA (500 mg/kg bwt, i.p.) followed by GTN (6 mg/kg bwt, i.p.) after 1 h of TAA administration. l-NAME was administered (40 mg/kg bwt, i.p.) after 1 h of TAA administration. ^†^ Significantly (*p* < 0.05) different from the saline-treated group. ^††^ Significantly (*p* < 0.001) different from the saline-treated group. * Significantly (*p* < 0.05) different from the TAA-treated group. ** Significantly (*p* < 0.01) different from the TAA-treated group. *** Significantly (*p* < 0.001) different from the TAA-treated group.

**Table 3 molecules-27-04362-t003:** Experimental design and group treatment in rats.

Group Number	Treatment Groups	Dosage Regimen
Group I, Control	Sodium chloride 0.9% *w*/*v* (Saline)	A dose of 10 mL/kg bwt, at the same time as the test agents. Control for test groups, II, III, and IV.
Group II	GTN	GTN: 6 mg/kg bwt, 1 h before killing the rats.
Group III	l-NAME	l-NAME: 40 mg/kg bwt, at the same times as l-NAME treatment in group VII.
Group IV	TAA	TAA: 500 mg/kg bwt, at the same time as TAA treatment in the groups V, VI, and VII.
Group V, Experimental Dose 1	TAA + GTN (3 mg/kg bwt)	TAA: 500 mg/kg bwt + GTN: 3 mg/kg bwt, 1 h after TAA administration.
Group VI, Experimental Dose 2	TAA + GTN (6 mg/kg bwt)	TAA: 500 mg/kg bwt + GTN: 6 mg/kg bwt, 1 h after TAA administration.
Group VII, Experimental	TAA + l-NAME	TAA: 500 mg/kg bwt + l-NAME: 40 mg/kg bwt, 1 h after TAA administration.

## Data Availability

Not applicable.

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
