# Peer review of "Is Glyceryl Trinitrate, a Nitric Oxide Donor Responsible for Ameliorating the Chemical-Induced Tissue Injury In Vivo?"

_molecules, 2022, doi:10.3390/molecules27144362_

Round 1
Reviewer 1 Report
This is an interesting manuscript detailing a study to determine if a nitric oxide donor molecule will paradoxically reverse ROS damage due to thioacetamide (TAA) exposure. All of the studies were carried out in old fashioned enzyme assays and basic histology, which is rarely seen in modern science. While I am not an expert on the exact approach and area of study, I do see areas for great improvement. First, the title is a question sentence without a question mark at the end. Also, the abstract has duplicative statements, and this is probably due to English not being the primary language of the communicating investigator.
Figure 1 should have the treatments labeled on the figures (bars).....this could be done by replacing Roman numerals with the treatment abbreviations. Stating what the data represent without details of treatments is confusing....but the authors list these below the Figure legend - very confusing. The same is true for Figure 2.
Histological figures are terribly messy, and not at all convincing of the conclusions. This part needs significant improvement.
In neither the Discussion nor the Conclusions do the authors answer the question posed by the title. The last sentence of the Conclusion make no sense!
Author Response
Reviewer # 1: Modifications made in revised manuscript in response to reviewer comments
We are pleased to learn that reviewer appreciated our work and gave positive and valuable comments to improve our manuscript. We have incorporated the comments and modifications as suggested. Changes are highlighted in the revised manuscript with red color. The details are as follows:
- Title : question mark added at the end of title as suggested
- Abstract: duplicative word has been corrected as suggested
- Figures 1 and 2: treatments details has been added as suggested
- Histological figures 3 A to F has been improved as suggested
- Discussion has been improved as suggested
- Conclusion has been rewritten and improved as suggested
We hope that revised manuscript will be acceptable for its publication to Molecules.
With regards,
Yours sincerely
Mohammad Iqbal, Ph.D.
Reviewer 2 Report
In the manuscript entitled “Is glyceryl trinitrate, a nitric oxide donor responsible for ameliorating the chemical induced tissue injury in vivo. Role of glyceryl trinitrate and thioacetamide in rat tissue injury” the authors evaluate the protective effect of glyceryl trinitrate liver injury caused by thioacetamide using in vivo models. Overall, the work is well done, carefully thought out, and performed, and the manuscript is well written and easy to read and follow. All experimental methods are well explained. The methods used are consistent with the literature and corroborate the objectives. The results presented are significant; however, they are very similar at present in the literature. The conclusions are supported by other data present in the literature. Other Specific comments:
Please insert the description of dose 1 and dose 2 on all tables and figures.
Please demonstrate the induced tumor promoter markers. No methodology is demonstrated to evaluate this objective.
Then, the authors must highlight the real contribution of this paper concerning that previously published. What is the contribution to the state of art?
Rahman, A., S. M. Vasenwala, and M. Iqbal. "Hepatoprotective potential of glyceryl trinitrate against chemically induced oxidative stress and hepatic injury in rats." Human & Experimental Toxicology 36.8 (2017): 785-794.
Al-Jawad, Faruk H., Zaid Al-Attar, and Muayyad S. Abbood. "The protective effect of nitroglycerin, N-acetyl cysteine and metoprolol in CCL4 induced animal model of acute liver injury." Open access Macedonian journal of medical sciences 7.11 (2019): 1739.
The authors need to demonstrate as endogenous production of NO reduces liver injury. This effect is correlated with the increase of antioxidant enzymes including glutathione peroxidase (GPx), glutathione reductase (GR), and glutathione-S-transferase (GST)?
Why the amount of nitrite not detected to TAA + GTN groups?
It is suggested that the authors proposed the possible mechanism of action involved in injury protection.
The quality of English writing throughout the manuscript needs adjusting. Spacing, punctuation marks, grammar, and spelling errors should be reviewed thoroughly. I found typos during the manuscript. Some sentences in the discussion section look unusual (Generally speaking). The whole manuscript must be evaluated by a native speaker or professional of English assistance may be required.
Author Response
Reviewer # 2: Modifications made in revised manuscript in response to reviewer comments
We are pleased to learn that reviewer appreciated our work and gave positive and valuable comments to improve our manuscript. We have incorporated the comments and modifications as suggested. Changes are highlighted in the revised manuscript with red color. The details are as follows:
- Description of dose 1 and dose 2 on all tables and figures has been added as suggested.
- We measured induced tumor promoter makers such as ornithine decarboxylase and [3H]thymidine incorporation in rat liver and have been demonstrated in abstract, materials and method (4.13. and 4.14.) and discussion section as suggested.
- Real contribution of this paper “endogenous production of NO from GTN to demonstrate that NO generated through the administration of GTN might scavenge ROS and decrease the toxic metabolite of TAA, thereby inhibiting hepatic oxidative stress, toxicity and tumor promotion in rats” has been highlighted and discussed in abstract, discussion and conclusion section as suggested.
- Based on the previous study of Rahman et al., Hepatoprotective potential of glyceryl trinitrate against chemically induced oxidative stress and hepatic injury in rats. Hum Exp Toxicol 2017; 8: 785-794, we only measured the amount of nitrite generated by glyceryl trinitrate (GTN) in vitro to demonstrate that endogenous NO generated through the administration of GTN might scavenge ROS and decrease the toxic metabolite of TAA in rats as a possible mechanism of action involved in liver injury protection. We did not measured nitrite in TAA + GTN groups.
We hope that revised manuscript will be acceptable for its publication to Molecules.
With regards,
Yours sincerely
Mohammad Iqbal, Ph.D.
Reviewer 3 Report
Manuscript ID: molecules-1743453
Title: Is glyceryl trinitrate, a nitric oxide donor responsible for ameliorating the chemical induced tissue injury in vivo. Role of glyceryl trinitrate and thioacetamide in rat tissue injury by Ayesha Rahman et al.
The authors aimed at the assessment of hepatoprotective action of glyceryl nitrate in TAA-induced liver injury. The overall quality of language, editorship and contextual logic should be improved. The main shortcomings are listed below.
Title - the title is too long. I should be more concise, for example “ Is glyceryl trinitrate, a nitric oxide donor responsible for ameliorating the thioacetamide- induced liver injury in rat.”
Abstract
1. “We study the effect of glyceryl trinitrate (GTN), a donor of nitric oxide and NG- nitroarginine methyl ester (L-NAME), a nitric oxide inhibitor along with these toxins. In the current study, the effect of TAA is studied in rat model with GTN and L-NAME. – repetition
2.“A group of seven healthy rats were selected for this study.”- only seven rats were used????
3.”… in the levels of both hepatic and serum enzymes including glutathione, malondialdehyde (MDA)” glutathione and MDA are not enzymes
4. “The treatment with GTN attenuated TAA-induced oxidative stress, liver injury and showed a strong inhibition of TAA induced tumor promotor markers” - this part of the sentence is not justified ; promotion of tumor is a process in which existing tumors are stimulated to grow (NCI definition)
Materials and methods
1. Lines 317-324 – this fragment is extremely unclear. It should be explained whether additional rats were used for the determination of ODC activity and 3H thymidine incorporation. If so, these rats should be shown in the table illustrating experimental protocol.
2. Page 9, Table – groups II, III and IV cannot be named “control” because they were administered chemical compounds. According to generally accepted principle control group are animals without treatment except for water or vehicle.
3. The title of the column in the table “Chemical with Intraperitoneal Injection” is wrong and should be changed to “Dosage regimen”.
4. It should be specified in what material (PMS, microsomes or homogenate) the described parameters were determined
Results
1.The title of figures is missing. In the legend the lack of explanation what parameters are shown in parts (a) and (b).
2. Lines 192- 195 “2.6. Amount of NO generated by GTN. The maximum amount of nitrite detected was found to be 0.75 ± 0.2 and 1.7 ± 0.3 mM/g tissue, at 3 and 6 mg doses of GTN, following incubation of GTN with hepatic PMS for a total of 45 min.” – This sentence is unclear. Does it mean that PMS of GTN-treated rats was incubated with GTN, what about controls? What is the rationale for such procedure? The analysis should be described in Materials and Methods. The title of 2.6. should be “Amount of nitrite generated by GTN”. Moreover, these data were not commented on in the Discussion
3. Results for G6PD are shown in Table1. It is discrepancy because this enzyme is not directly involved in GSH metabolism
Discussion
1.Some statements were not supported by cited references. Examples:
Lines 215-216 “NO is demonstrated to inhibit oxidant generation [3]…” – no inhibition of oxidants generation has been evidenced in [3].
Lines 250-252 “… showing the beneficial role of GTN as an anti-proliferative agent by reducing synthesis of the protein and DNA in corporation are supported with these results [3, 5, 26, 49]. In the articles [3], [26],[49] antiproliferative properties of GTN were not assayed
Lines 238-239 “L-NAME act as an oxidant and that was helpful in the supporting the protective role of GTN in this study [3, 42].” Pro-oxidant properties of L-NAME have not been demonstrated in the article [3], in [42] L-NAME is not mentioned at all.
2.The interpretation of the results related to antioxidant status parameters is unintelligible and not convincing: lines 231-238 “…which could be as a result of TAA-induced persistent oxidative stress […] oxidant and that was helpful in the supporting the protective role of GTN in this study [3, 42].”
3.Lines 265-269 “Our study supports this model [….] lessening the cell death cycles and fibrotic regeneration that are characteristic of the cirrhotic process.”
This statement is not justified because neither cell cycle nor fibrotic regeneration were examined.
Terminology errors
-The term “glutathione-metabolizing enzymes” is not correct. “GSH-dependent enzymes”
is a proper one and is generally accepted in scientific reports
-“redox cycling enzymes” – there is no category of “redox cycling enzymes” because chemicals ( e.g. chinones or paraquat) undergo the process of redox cycling not enzymes
-“serological enzymes” – the correct term is “serum enzymes”
Style, grammar and syntax errors - examples
1. Lines 73-75 “…glyceryl trinitrate (GTN) exert the healing effect on the injury along with its healing effect if at all on the stress produced by the oxidants along with the single necrogenic TAA dose.” Which oxidants are mentioned? TAA alone was used in the experiment.
2. Lines 106-109 “Table 2 shows the GTN administration effect on TAA-mediated serum transaminase release is reported in Table 2. A significant (p<0.001) release of the marker enzymes (alanine aminotransferase (AST) and aspartate aminotransferase (ALT)) of damage to the tissue membrane in serum were notice with TAA administration.” Grammar and syntax errors
3. Line 166 “2.5. Effect of GTN and L-NAME on TAA-induced hepatic histopathological studies” Effect of GTN on studies?
Author Response
Reviewer # 3: Modifications made in revised manuscript in response to reviewer comments
We have incorporated the comments and modifications as suggested. Changes are highlighted in the revised manuscript with red color. The details are as follows:
Title- modified as suggested.
Abstract
- “We study…………….repetition” has been corrected in abstract.
- “Seven groups of healthy rats were selected for this study. Only seven rats were used?” sentence has been corrected in abstract.
- “…In the levels of both hepatic and serum enzymes…….. ……glutathione and MDA are not enzymes” sentence has been corrected in abstract.
- “The treatment with GTN attenuated……………..induced tumor promoter markers” sentence has been corrected in abstract.
Materials and methods
- Lines 317-324- Sentence has been rewritten and for [3H]thymidine incorporation study, there were additional forty-two male Wistar rats were used.
- Page 9, Table-group II, III and IV cannot be named “control” has been corrected as suggested.
- Title of the column in the table has been changed to dosage regimen as suggested.
- In what materials (PMS, microsomes or homogenate) the described parameters were determined has been mentioned as suggested.
Results
- Title of figures is added and explanation of parameters is given.
- Line 192-195 “Title amount of nitrite generated by GTN” corrected as suggested.
- Results for G6PD corrected as suggested.
Discussion
- Lines 215-252 Correct reference cited as suggested.
- Lines 250-252 We measured markers of tumor promoters (hyper proliferative response) such as ODC and [3H]thymidine suggesting antiproliferative properties of GTN.
- Lines 238-239 “L-NAME acts as an oxidant…….in this study”. We have taken group II (L-NAME) and group VII (TAA + L-NAME). L-NAME increased the toxicity of TAA as observed in this study suggesting that it act as oxidant.
- Interpretation of the results related to antioxidant status has been rewritten.
- Lines 265-269 “Our study supports….cirrhotic process” sentence has been modified as suggested.
Terminology errors
- The term “glutathione metabolizing enzymes” has been replaced by “GSH-dependent enzymes” as suggested.
- “redox cycling enzymes” has been replaced by “redox cycling” as suggested.
- “Serological enzymes” has been replaced by “serum enzymes” as suggested.
Style, grammar and syntax errors-example
- Line 73-75 “…glyceryl trinitrate (GTN) exert…….TAA dose” sentence has been corrected.
- Lines 106-109 “Table 2 shows…………..TAA administration” sentence has been corrected.
- Line 166 “2.5. Effect of GTN and L-NAME histopathological studies” has been corrected.
We hope that revised manuscript will be acceptable for its publication to Molecules.
With regards,
Yours sincerely
Mohammad Iqbal, Ph.D.
Round 2
Reviewer 1 Report
The revised manuscript addressed my concerns and suggestions
Author Response
I am happy to know that reviewer recomended paper for its publication to molecules.
Thanks
With Regrads
Mohammad Iqbal, PhD
Reviewer 2 Report
No more comments.
Author Response

(The authors gave the same response as above.)

Reviewer 3 Report
MOLECULES Glyceryl trinitrate review II
The manuscript editing has not been sufficiently improved. There are still a lot of terminology, style and syntax errors which render understanding of the text difficult. In the previous review I indicated only examples, I did not provide a complete correction of the text - it is not a role of a reviewer. The manuscript should be corrected by the native speaker familiar with life sciences.
The Authors ignored several remarks from my first review:
I. In their response the Authors wrote: Results for G6PD corrected as suggested. It is not true because in the new version of the manuscript no change has been introduced.
II. There is no response to my remark from the previous review : Lines 250-252 “… showing the beneficial role of GTN as an anti-proliferative agent by reducing synthesis of the protein and DNA in corporation are supported with these results [3, 5, 26, 49]. In the articles [3], [26],[49] antiproliferative properties of GTN were not assayed
III. The Authors used again the term “tumor promoter markers” – the correct term is “tumor promotion markers”
Other amendments:
1. The conclusion in the last sentence is illogical. The decrease in the activity of serum enzymes does not prove that GTN is an antioxidant.
2.Figure titles should be changed: 1a. Effect of GTN and L-NAME on hepatic GSH in rats treated with TAA; 1b. Effect of GTN and L-NAME on lipid peroxidation in rats treated with TAA; 2a. . Effect of GTN and L-NAME on hepatic ODC activity in rats treated with TAA;
2b. Effect of GTN and L-NAME on 3H thymidine incorporation in rats treated with TAA
Fig 3. The title is missing
3. Additional examples of errors are given below
- The Authors use the term “tumor promoter markers” – the correct term is “tumor promotion markers”
-Lines 116-118 – “However, the damage caused by subsequent serum AST and ALT was further enhanced…” illogical, ALT and AST cannot cause any damage
- Lines 142-142 - …improvement in LPO were notice noticed when given NO inhibitor L-NAME with TAA when compared to the TAA-treated group. – repetition
- Lines 142-143 GTN and L-NAME alone did not indicate cause any major changes in those parameters.
- Lines 171-172- TAA administration showed increase caused the increased infiltration of the inflammatory cellular components
- Lines 207-208 – “…as the main issue with NO remains being it a pro-oxidant at high concentration.” This phrase is awkward - change or delete
- Lines 226-227 – “ GTN exert the protection in liver injury that was evident by the amelioration of the enzyme levels.” The enzymes should be specified.
- Lines 226-227 – “The improvement of GTN mediation on TAA induced oxidative stress may be due to the NO-generated antioxidant titer of TAA oxidants”. Illogical and unintelligible. The term “titer” refers to antibodies and is not correct in this context
-Line 247 – “The attenuation of GTN-mediated GSH dependent enzymes… - incorrect terminology
Line 250-251 - L-NAME act as an oxidant and GTN was helpful in protecting hepatic cells against TAA induced injury as observed in this study. Why L-NAME is mentioned in this sentence if the statement refers to GTN effect on TAA induced injury?
-Line 256 – “TAA’s death of a cell in necrosis….” Wrong term “TAA’s death”
-Line 273 - “… indicated an induction in ALT, GGT and AST levels…” – transaminase assays do not show induction of these enzymes but their release from the liver to serum
Author Response
Reviewer # 3: Modifications made in revised manuscript in response to reviewer comments
We have incorporated the comments and modifications as suggested. Changes are highlighted in the revised manuscript with red color. The details are as follows:
Terminology, style and syntax errors have been improved in whole manuscript as suggested.
- Results of G6PD have been discussed in discussion section as suggested.
- Line 250-252 “showing beneficial role of GTN……with these results [3,5,26,49]” sentence has been modified in discussion section and new reference has been added as suggested.
III. The term “tumor promoter markers” has been replaced with correct term “tumor promotion markers” in whole manuscript as suggested.
Other amendments:
- The conclusion in the last sentence has been modified as suggested. “The decrease……is an antioxidant”.
- Figure titles have been changed in 1a, 1b, 2a, 2b as suggested and figure 3 title has been included as suggested.
- Additional examples of errors have been corrected as suggested are given below:
- Term “tumor promoter markers” has been replaced with correct term “tumor promotion markers” in whole manuscript as suggested.
- Lines 116-118 “However, the damage….further enhanced” sentence has been modified as suggested.
- Lines 142-142 “improvement in LPO…TAA treated group” corrected as suggested.
- Lines 142-143 “GTN and LNAME alone….parameters” corrected as suggested.
- Lines 171-172 “TAA administration showed….cellular components” corrected as suggested.
- Lines 207-208 “as the main issues with NO…….high concentration” sentence deleted as suggested.
- Lines 226-227 “GTN exert the protection in liver injury….enzyme levels” name of enzymes specified as suggested.
- Lines 226-227 “The improvement of GTN…. TAA oxidants” sentence corrected as suggested.
- Line 247 “The attenuation of GTN……..GSH dependent enzymes” terminology sentence corrected as suggested.
- Line 250-251 “L-NAME act as an ……….TAA induced injury” sentence corrected as suggested.
- Line 256 “TAA’s death---cell in necrosis” sentence corrected as suggested.
- Line 273 “indicated an induction……..levels” sentence corrected as suggested.
We hope that revised manuscript will be acceptable for its publication to Molecules.
With regards,
Yours sincerely
Mohammad Iqbal, Ph.D.